# MULTI-AGENT HIERARCHICAL REINFORCEMENT LEARNING FOR HUMANOID NAVIGATION

## ABSTRACT

Multi-agent reinforcement learning is a particularly challenging problem. Current methods have made progress on cooperative and competitive environments with particle-based agents using communication and centralized training. Little progress has been made on solutions that could operate in the real world with interaction, dynamics, and humanoid navigation strategies. In this work, we make a significant step in multi-agent models on simulated humanoid navigation by combining Multi-Agent Reinforcement Learning with Hierarchical Reinforcement Learning. Specifically, we develop a partial parameter sharing approach wherein the lower level of the hierarchy is shared enabling learning using decentralized methods. This drastically reduces the overall parameter space in the multi-agent problem and introduces structure in the optimization problem.

We build on top of prior foundational work in learning goal-conditioned policies to learn low-level physical controllers for balance and walking. These lower-level controllers are task-agnostic and can be shared by higher-level policies. Overtop of these goal conditioned policies, we can train decentralized heterogeneous policies for multi-agent goal-directed collision avoidance. Surprisingly, our results show that with this combination of methods, RL techniques can be used for finding strong policies. A video of our results on a multi-agent pursuit environment can be seen here[1].

## 1 INTRODUCTION

Deep Reinforcement Learning (DRL) has been successful in solving complex planning tasks. Given a sizeable computational budget DRL has displayed superhuman performance on many games (Mnih et al., 2015; Silver et al., 2017). However, less progress has been made on the Multi-Agent Reinforcement Learning (MARL) problem space, possibly due to the non-stationary optimization of multiple changing policies, which is not easily overcome by collecting more data (OpenAI, 2018). If the goal is to create agents that can operate in a dynamic multi-agent world, more stable methods with novel forms of communication are needed. The trend to make progress on MARL has been to simplify the optimization problem. For example, converting the multi-agent problem into a single agent centralized model results in large gains in performance but can increase the number of network parameters significantly and imposes a constraint on the generalization to the number of agents (Lowe et al., 2017). By using recurrent policies, significant computational power, and constraints on the amount the policy is allowed to change between updates, it is possible to beat the best humans at the multi-agent game of Dota (OpenAI, 2018). While these methods have shown promise, they require significant amounts of compute and have not yet displayed success in complex and dynamic multi-agent environments. While recent work has been successful in producing competitive behaviour using asymmetric self-play, this work is limited to two agents (Bansal et al., 2017).

In this work, we propose the integration of MARL with Hierarchical Reinforcement Learning (HRL) to produce heterogeneous humanoid agents that can both navigate and interact in dynamic simulation. Specifically, we propose a method to reduce the complexity in the MARL policy optimization problem using a type of parameter sharing. While previous methods have focused on re-framing the problem as a type of single-agent RL problem, we show that this does not scale. Our method

---

[1] https://sites.google.com/view/mahrl

preserves the essential features of heterogeneous agent behaviour without adding more network parameters (Lowe et al., 2017). This use of HRL with a goal conditioned lower layer (Kaelbling, 1993), has significant advantages over current methods. We use shared parameter methods for task-agnostic portions of the policy similar to multi-task learning (Caruana, 1997; Argyriou et al., 2007; Lu et al., 2017) where each high-level agent is a different task. Given the shared sub-policy, the optimization is simplified and allows learning complex multi-agent policies with significantly less data. This method represents, to the best of our knowledge, the first method for heterogeneous multi-agent physical character control for locomotion, navigation, and behaviour.

## 2 RELATED WORK

Simulated robot and physical character control is a rich area of research with many solutions and approaches. Neural models focused on training neural networks by receiving joint or body sensor feedback as input and producing appropriate joint angles as output (Geng et al., 2006; Kun & Miller III, 1996; Miller III, 1994). A biped character's movement controller set can also be manually composed using simple control strategies and feedback learning (Yin et al., 2007; Faloutsos et al., 2001). HRL has been proposed as a solution to handling many of the issues with current RL techniques that have trouble with long horizons and weak signal. Many frameworks have been proposed for HRL, but none seem to be the obvious choice for any particular problem (Sutton et al., 1999; Dayan & Hinton, 1993; Dietterich, 1999). One difficulty in HRL design is finding a reasonable communication representation to condition the lower level. Some methods pretrain the lower level on a random distribution (Heess et al., 2016; Peng et al., 2017; Merel et al., 2018) and others learn a more constructive latent encoding (Nair et al., 2018; Eysenbach et al., 2019; Gupta et al., 2018). There is also the present challenge of learning multiple levels of the hierarchy concurrently (Vezhnevets et al., 2017; Nachum et al., 2018; Levy et al., 2017a). Here a goal-based approach that uses a footstep space representation that is pretrained to learn a task-agnostic lower level of control. This provides the high-level policies with a single shared model that is conditioned to produce diverse behaviour based on a latent goal input.

### 2.1 MULTI-AGENT DEEP REINFORCEMENT LEARNING

There are many types of multi-agent learning problems, including cooperative-competitive and with-without communication (Bu et al., 2008; Tan, 1993; Panait & Luke, 2005). Recent work converts the MARL problem to a single agent setting by using a single Q-function across all agents (Lowe et al., 2017). Additional work focuses on the problem of learning communication methods between agents (Foerster et al., 2016). While progress is being made, MARL is notoriously tricky due to the non-stationary optimization issue, even in the cooperative case (Claus & Boutilier, 1998). In this work, we apply a partial *parameter sharing* method assuming all agents share task-agnostic locomotion and optimize similar goals (Gupta et al., 2017).

There exists few environments specifically created for MARL evaluation (Zheng et al., 2018; Suarez et al., 2019). The focus of these environments is often a type of strategy learning and cooperation (Tian et al., 2017; Vinyals et al., 2017). MARL is a growing area of research and, as such, will need increasingly complex environments to evaluate algorithms. In this work, we are interested in the overlapping problems of control and perception. To this end, we have created the first simulation of its type that affords multiple physics-based control tasks with variable numbers of agents. Recent work has begun to combine MARL and HRL but is limited to simple environments, uses additional methods to stabilize the optimization, and includes communication (Tang et al., 2018; Han et al., 2019). This work focuses on a decentralized approach for interacting models that generalize across scenarios, agent numbers, and tasks.

Our work represents a new paradigm in learning within the multi-agent navigation domain that goes beyond simple navigation control strategies. We show compelling AI that learns navigation and gameplay strategies with fully articulated physical characters. This is achieved through a novel learning strategy that produces high-value policies for a complicated control problem.

## 3    POLICY REPRESENTATION AND LEARNING

In this section, we outline key details of the general Reinforcement Learning (RL) framework. RL is formulated on the Markov Dynamic Process (MDP) framework: at every time step $t$, the world (including the agent) exists in a state $s_t \in S$, wherein the agent is able to perform actions $a_t \in A$, sampled from a policy $\pi(s_t, a_t)$ which results in a new state $s_{t+1} \in S$ according to the transition probability function $T(s_{t+1}|a_t, s_t)$. Performing action $a_t$ in state $s_t$ produces a reward $r_t$ from the environment; the expected future discounted reward from executing a policy $\pi$ is:

$$J(\pi) = \mathbb{E}_{r_0,\ldots,r_T} \left[ \sum_{t=0}^{T} \gamma^t r_t \right] \tag{1}$$

where $T$ is the maximum time horizon, and $\gamma$ is the discount factor, indicating the planning horizon length. The agent's goal is to optimize its policy, $\pi(\cdot|\theta_\pi)$, by maximizing $J(\pi)$. The policies in the work are trained using the Proximal Policy Optimization (PPO) algorithm (Schulman et al., 2017). The value function is trained using TD($\lambda$).

### 3.1    HIERARCHICAL REINFORCEMENT LEARNING

In HRL the original MDP is separated into different MDPs that should be easier to solve separately. In practice, this is accomplished by learning two different policies in two different layers. The lower level policy is trained first and is often conditioned on a latent variable or goal $g$. The lower level policy $\pi^{lo}(a|s, g)$ is constructed in a way to give it temporally correlated behaviour depending on the $g$. After the lower level policy is trained, it is used to help solve the original MDP using a separate policy $\pi^{hi}(g|s)$. This policy learns to provide goals to the lower policy to maximize rewards.

## 4    MULTI-AGENT REINFORCEMENT LEARNING

The extension to the MDP framework for MARL is a partially observable Markov game (Littman, 1994). A Markov game has a collection of $N$ agents, each with its own set of actions $A_0, \ldots, A_N$ and partial observations $O_0, \ldots, O_N$ of the full state space $S$. Each agent $i$ has its own policy $\pi(a|o_i, \theta_i)$ that models the probability of selecting an action. The environment transition function is a function of the full state and each agent's actions $T(S'|S, A_0, \ldots, A_N)$. Each agent $i$ receives a reward $r_i$ for taking a particular action $a_i$ given a partial observation $o_i$ and its objective is to maximize this reward over time $\sum_{t=0}^{T} \gamma^t r_i^t$ , where $\gamma$ is the discount factor and $T$ is the time horizon. The policy gradient can be computed for each agent as

$$\nabla_{\theta_i} J(\pi(\cdot|\theta_i)) = \int_{O_i} d_{\theta_i}(o_i) \int_{A_i} \nabla_{\theta_i} \log(\pi(a_i|o_i, \theta_i)) A_\pi(o_i, a_i) \, da_i \, do_i \tag{2}$$

where $d_\theta = \int_S \sum_{t=0}^{T} \gamma^t p_0(o_0)(o_0 \to o \mid t, \pi_0) \, do_0$ is the discounted state distribution, $p_0(o)$ represents the initial state distribution, and $p_0(o_0)(o_0 \to o \mid t, \pi_0)$ models the likelihood of reaching state $s$ by starting at state $o_0$ and following the policy $\pi(a, o|\theta_\pi)$ for $T$ steps (Silver et al., 2014). Here $A_\pi(o, a)$ represents the advantage function estimator GAE($\lambda$) (Schulman et al., 2016).

The challenge in MARL is that each agent learns separately, and this often results in a non-stationary learning problem. Each agent is learning how to estimate the dynamics and advantage of an environment that is affected by other actively learning agents. Data collected and used for learning becomes inaccurate after any learning update; thus, the problem is non-stationary. MADDPG reformulates the above problem into a fully observable problem with a centralized Q-function using the equation:

$$\nabla_{\boldsymbol{\theta}_i} J(\boldsymbol{\theta}) = \mathbb{E}_{o_i \sim p_\pi(\cdot), a_i \sim \pi(\cdot|o_i)}[\nabla_{\boldsymbol{\theta}_i} \log \pi(a_i|o_i, \theta_i) Q_i^\pi(\mathbf{o}, \mathbf{a})] \tag{3}$$

Here $Q_i^\pi(\mathbf{o}, \mathbf{a})$ is a *centralized* action-valued function that gives the Q value for agent $i$. Where $\mathbf{o} = \{o_t^0, \ldots, o_t^N\}$ is the observations of all agents and $\mathbf{a} = \{a_t^0, \ldots, a_t^N\}$ all agent actions at time $t$ and $\boldsymbol{\theta} = \{\theta_0, \ldots, \theta_N\}$ is the collection of policy parameters for each agent. This framework will allow the method to train the policies together, but it treats the agents as disjoint distributions to optimize. Instead, we propose a *decentralized* method that does not need to learn a complex Q-function and reduces the optimization challenges by using HRL.

### 4.1 TASK-AGNOSTIC LOCOMOTION CONTROLLER (LC)

The LC, the lower-level policy in our design, is designed to learn a robust and diverse policy $\pi(a_t|o_t, g_t, \theta_t^{lo})$ based on a latent goal $g_t$ varaible. The footstep goals $g_L = \{\hat{p}_0, \hat{p}_1, \hat{\theta}_{root}\}$ in Figure 1, consist of the agent root relative distances of the next two footsteps on the ground plane and the desired facing direction at the first step's end. This goal description is motivated by work that shows people may plan steering decisions two foot placements ahead (Zaytsev et al., 2015).

The LC learns to place its feet as accurately as it can to match these step goals using the reward $r_{Lg} = \exp(-0.2||\mathbf{s}_{char}^g - g_L||^2)$. The better the LC learns this behaviour, the more responsive the controller will be to provided goals.

### 4.2 MULTI-AGENT HIERARCHICAL REINFORCEMENT LEARNING

We construct a multi-agent learning structure that takes advantage of hierarchical design or Multi-Agent Hierarchical Reinforcement Learning (MAHRL). Each agent has its own higher level policy (Multi-Agent Navigation Controller (NC)) $\pi(g|o, \theta_i^{hi})$ and a shared task agnositic lower level policy (LC) $\pi(a|o, g, \theta^{lo})$. This method allows us to introduce more structure into the difficlut multi-agent optimization problem. This change alters the underlying MDP, such that the policy is queried for a new action every $k$ timesteps. This also changes the MDP method by reducing the dimensoinality of the action space to specifiying goals $g$ while using the low-level policy to produce more temporally consistent behaviour in the original action space. The low-level policy is used in a deterministic manner to futher reduce variance introduced into the problem. We will use the notation $\pi(a_i|o_i, \theta_i^{hi}, \theta^{lo})$ to denote the policy induced by the pair of policies. While current research shows that it is challenging but possible to train a two-level hierarchy concurrently (Nachum et al., 2018; Levy et al., 2017b), we instead pretrain the lower level policy and leave it fixed and shared across all agents. This reduces the MARL problem from learning the details of locomotion via joint torques for each agent to learning goal-based footstep plans for each agent.

The use of HRL is key to the method. When the challenge in MARL is dealing with what can be large changes in the distribution of states visited by the agent, the use of a temporally correlated structure given by the goal-conditioned LC significantly reduces the non-stationarity. Not only is each agent sharing its network parameters with each other agent, but this portion has also been carefully constructed to provide structured exploration for the task, thus greatly reducing the number of network parameters that need to be learned. This is in contrast to *centralized* methods that take a step away from the goal of solving the heterogeneous problem in a scalable way. We extend the analysis of this work by combining the PPO algorithm in a MARL *partial parameter sharing* setting. Details of the algorithm can be found in the supplementary material. The use of the LC controls the way $d_{\theta_i}(o_i)$ can change for each agent making it easier for each agent to estimate other agents potential changes in behaviour because the LC is already trained to produce a specific type of useful behaviour that is a subset of the full space. We also create a version of MADDPG that uses the HRL structure and show that it does not perform as well.

## 5 LEARNING HIERARCHICAL MULTI-AGENT NAVIGATION CONTROL

To solve the hierarchical learning problem, we train in a bottom-up fashion sharing the LC policy among heterogenous NC policies. The levels of the hierarchy communicate through shared actions and state in a feedback loop that is meant to reflect human locomotion. The NC's objective is to provide footstep placement goals $a_H = g_L$ for the LC as seen in the right hand side of Figure 1. These footstep goals are produced as two-step plans. Each step is parameterized with its root-relative placement, angle on the ground, centre of mass heading vector, and a time signature. The NCs are queried for a new footstep action every 0.5 s. The NCs decides what action to take based on the egocentric velocity field $E$ in front of the agent, its pose $\mathbf{s}_{char}$ and the NC goal $g_H$, which together form the NC state $C = \{E, \mathbf{s}_{char}, g_H\}$, seen in the left box of Figure 1.

### 5.1 STATE SPACE

In several studies, it has been shown that optic flow, an inherently egocentric phenomenon of mapping velocities to regions of vision, is key to sighted locomotion (Bruggeman et al., 2007; Warren Jr

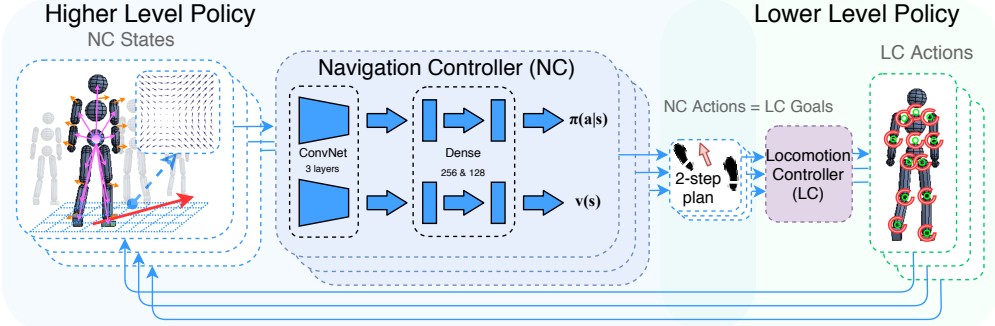

Figure 1: An overview of the MAHRL approach. From left to right: the NC state includes the relative goal position and distance, an egocentric velocity field capturing the relative velocity of obstacles and other agents, and the physical character link positions and linear velocities; for each agent this state is input to a multi-layer CNN, including two dense hidden layers, and outputs actions–the value function uses a similar network. These high-level actions are in the form of two-step plans dictating future foot placements; the LC consumes these footstep plans as $g$, which produces the angle-axis targets for joint PD controllers.

et al., 2001). Additionally, the visual field directly impacts walking ability and locomotion stability (Jansen et al., 2011). Taken together, vision's role in locomotion forms an egocentric velocity field where perceived distance and movement play different roles in locomotion control (Turano et al., 2005). This evidence has been used previously to define multi-agent navigation models, both by constructing a discretized egocentric field (Kapadia et al., 2009) and by learning the discretization of an egocentric field (Long et al., 2017).

The NC uses as input an egocentric velocity field relative to the agent's location and rotation. This egocentric velocity field $E$ is $32 \times 32$ samples over a space of 5x5 m, starting $0.5$ m behind the agent and extending $4.5$ m in front, shown in the left hand side of Figure 1. The velocity field consists of two image channels in the x and y directions of a square area directly in front of the agent, where each sample is calculated as the velocity relative to the agent (Bruggeman et al., 2007; Warren Jr et al., 2001). The current pose of the agent is included next, followed by the NC goal. The NC goal $g_H$ consists of two values, the agent relative direction and the distance to the current spatial goal location.

## 5.2 Simulation Environment & Training

We construct a collection of physics-based simulation tasks to evaluate the method. At initialization, each agent is randomly rotated, and the initial velocities of the agent's links are determined by a selected motion capture clip using finite differences and rotated to align with the agent's reference frame. Goal locations are randomly generated in locations that are at least 1 m away from any obstacle. Each agent is randomly placed in the scene such that it does not intersect with any other agent or obstacle. The number and density of agents in the simulation vary depending on the task. For training, we found that starting with 3 agents in the environment is a good trade-off between computation cost and the generalization ability of the resulting learned policy. Environment specifics are given in Table 1. We consider the simulation and training environment to be, to the best of our knowledge, another novel contribution. While some simulators exist that support physics-based simulation for robots (Brockman et al., 2016; Tassa et al., 2018), few support more than one agent, with at most 2 which interact beyond physical simulation and actually solve their tasks with respect to each other. Other libraries focus on supporting different kinds of MARL configurations for particle-based agents (Lowe et al., 2017). Our proposed approach represents the first physics-based simulation of its kind that supports MARL.

The NC uses convolutional layers followed by dense layers. The particular network used is as follows: 16 convolutional filters of size $6 \times 6$ and stride $2 \times 2$, 16 convolutional filters of size $3 \times 3$ and stride $1 \times 1$, the structure is flattened and the character and goal features $\mathbf{s}_{char}, g_H$ are concatenated, a dense layer of 256 units and a dense layer of 128 units are used at the end. The network uses Rectified Linear Unit (ReLU) activations throughout except for after the last layer which uses a *tanh* activation that outputs values between $[-1, 1]$. All network inputs are standardized with respect

| Name | agent count | obstacle count | size | agent direction | obs location |
|---|---|---|---|---|---|
| *Pursuit* | 3 | $[0, 10]$ | $10 \times 10$ m | random | random |
| *Soccer* | 4 | 0 | $10 \times 10$ m | towards centre $+\mathcal{N}(0, 0.1)$ | none |
| *Mall* | $[3, 5]$ | $[0, 10]$ | $10 \times 10$ m | random | random |
| *Bottleneck* | $[3, 5]$ | 4 | $10 \times 20$ m | right $+\mathcal{N}(0, 0.15)$ | around |

Table 1: Scenarios and their main parameters.

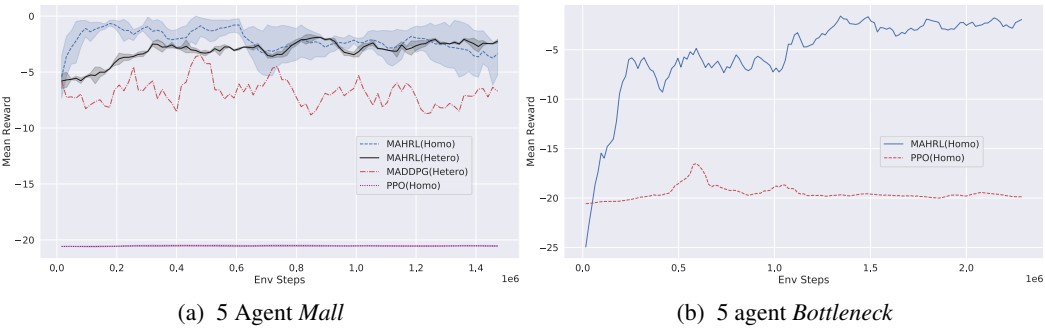

(a) 5 Agent *Mall*  (b) 5 agent *Bottleneck*

Figure 2: Comparative study of the learning curves of MAHRL, MADDPG, and PPO in the Mall scenario.

to all states observed so far. The rewards are scaled by a running variance. That is, the variance is computed from a running computation during training that is updated after every data collection step. The batch size used for PPO is 256, with a smaller batch size of 64 for the value function. The policy learning-rate and value function learning-rate are $0.0001$ and $0.001$, respectively. The value function is updated four times for every policy update. The NC also uses the Adam stochastic gradient optimization method (Kingma & Ba, 2014) to train the Artificial Neural Network (ANN) parameters.

## 6 RESULTS

In this section, we demonstrate the efficacy of MAHRL. We introduce a new set of challenging multi-agent simulations and show that the previous method can not solve these tasks. However, using MAHRL many of these tasks can be solved with a modest amount of data and computational power. We separate our evaluation into four sections. We examine the performance of MAHRL in terms of training, quantitative metrics, and learned policies. Then we examine the performance in terms of computation cost and generalizability over the number of agents in the environment.

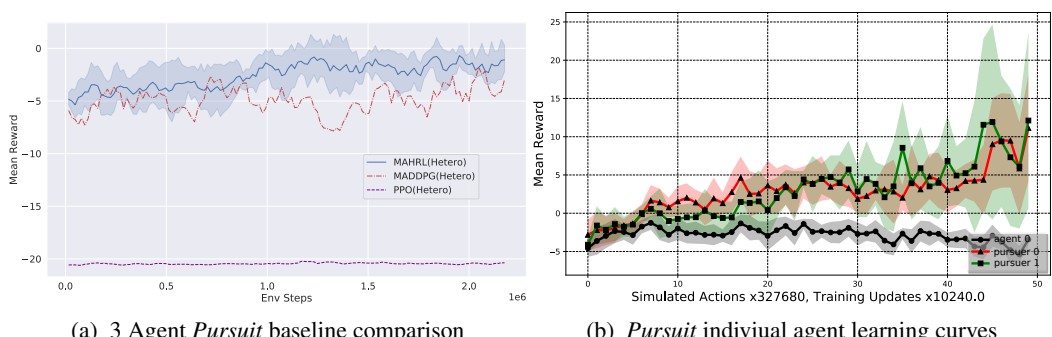

(a) 3 Agent *Pursuit* baseline comparison  (b) *Pursuit* indiviual agent learning curves

Figure 3: Comparative study of the learning curves of MAHRL, MADDPG, and PPO in the Pursuit scenario.

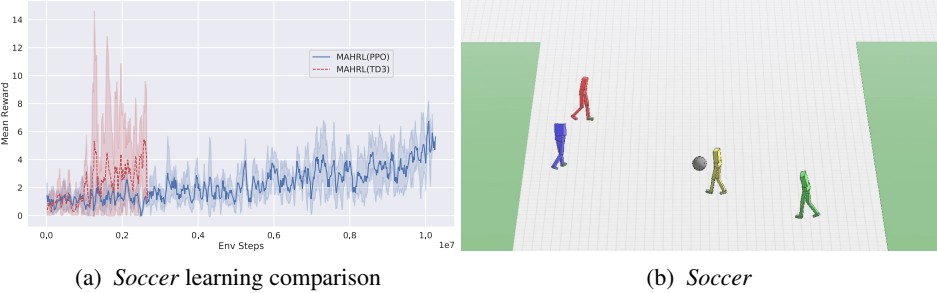

(a) *Soccer* learning comparison        (b) *Soccer*

Figure 4: Comparative study of the learning curves of MAHRL based on PPO (blue) and MAHRL using TD3 (red) in the *Soccer* scenario. The yellow and green agents are the same team, while the blue and red on the other team.

## 6.1 LEARNING

We evaluate MAHRL in complex multi-agent environments by first examining the performance of the reward function over training episodes. We show that MAHRL performs much better than the basic PPO algorithm without any hierarchical structure in the 5-agent *Mall* in Figure 2a. After a lengthy training session, the basic PPO is not able to even produce a standing behaviour. We find that MAHRL performs well and is able to learn navigation behaviour after a short amount of training. We find that the hierarchical design provides a significant improvement in this case. We also compare to MADDPG in Figure 2a,3a and find that, while it does better than PPO as it also uses the HRL structure, it struggles to produce good coordinated behaviour. We believe this is related to the large Q-network that needs to be learned for MADDPG. Qualitatively, throughout training, even with the increased control complexity, our method is able to learn successful navigation strategies shown in Figure 8. Each agent in the scenario quickly develops strong navigation behaviours that become more conservative over time as agents value avoiding collisions. This can result in agents taking longer paths in the environment.

MAHRL, is also applied to a *Pursuit* environment. In this environment, there is one agent (agent 0) with the same navigation goal as in previous environments. As well, two additional agents (pursuer 0,1) that have the goal of reaching agent 0. This is accomplished by setting a high-level goal $g_H$ for the pursuers to the location of agent 0. In Figure 3a, we show a comparative analysis of the learning curves of MAHRL, MADDPG, and PPO. We note that qualitatively, the three agents all begin to increase their average reward via their navigation objective, as learning progresses the pursuing agents outperform agent 0, shown in Figure 3b and as they get better agent 0 has an increasingly difficult time reaching its own navigation targets. An example of this behaviour is shown in Figure 7.

Last MAHRL is applied to a 2-on-2 soccer simulation, *Soccer* shown in Figure 4b. In this simulation, there are two teams of 2 bipedal agents. The goal of the members of each team is to kick the ball into the other team's goal area (shown in green). The task is particularly challenging to learn as each agent now has to learn with one cooperative agent and two adversarial agents. We find that MAHRL works well at learning policies for agents to locate and kick the ball into the opposing team's goal. The learning curves for two different methods that use MAHRL are shown in Figure 4a, on based on PPO and another based on TD3. Our results often produce imbalanced teams where one team becomes much better at the game than the other. This can be seen in the accompanying video on the website.

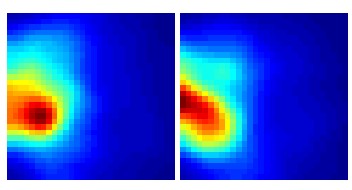

(a) ego field x    (b) ego field y

By extracting the gradients on the input features for the value function in the learned model, we can examine some artifacts of what is learned. Recall that for input, we include in the state of the NC a simple model of an egocentric perceptual field– a rectangular region in front of the agent. We show that our models learns two important aspects of navigation with respect to this field. The magnitude of the velocity field gradients in our learned model reveals that the learning process has developed an egocentric velocity field attenuated with distance. This can be seen in Figure 5a. Interestingly,

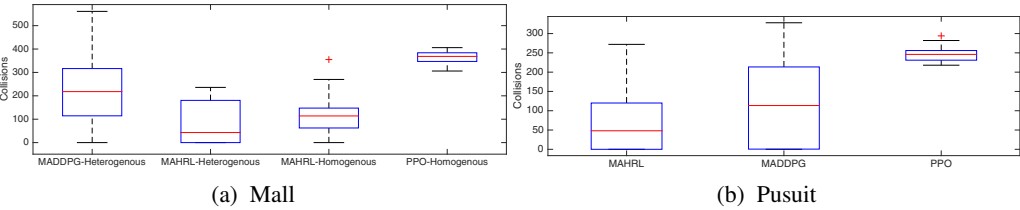

(a) Mall        (b) Pusuit

Figure 6: Comparative analysis of collisions counts across all baselines, MADDPG, MAHRL with and without heterogeneous agents, and PPO in the pursuit/tag scenario. MAHRL outperforms both MADDPG and PPO. In this game, the collisions are indicators of poor policies leading to negative collisions during pursuit and evasion.

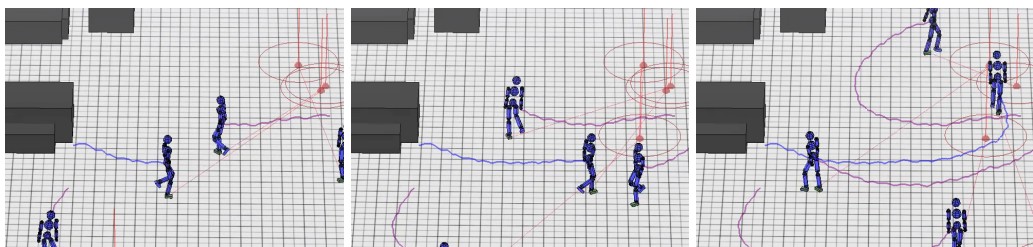

Figure 8: Rasterized images from the mall environment with humanoid agents navigating and avoiding each other while seeking goals. A video for this example can be found here.

this field is biased toward the rightward direction (down in the figures). This bias supports reciprocal collision avoidance in counter flows.

## 6.2 QUANTITATIVE COMPARISON

To evaluate learned navigation strategies quantitatively, we capture the mean number of collision events overall agents for each episode in several instantiations of the Mall environment. We perform this study over the state-of-the-art MADDPG method, MAHRL with and without Heterogeneous agents, and PPO. The results are shown in Figure 6a. For each model, we perform 155 policy rollouts over several random seeds. The basic PPO algorithm does not appear to learn anything. A Kruskal-Wallis rank-sum test and post-hoc Conover's test with both FDR and Holm corrections for ties show the MAHRL methods significantly outperform others (p ¡ 0.01).

To evaluate the pursuit and evade strategies quantitatively, we capture the mean number of collision events overall agents for each episode in several instantiations of the game. We perform this study over the MADDPG method, MAHRL, and PPO. The results can be seen in Figure 6. For each model, we run 155 simulations over several random seeds. A Kruskal-Wallis rank-sum test and post-hoc Conover's test with both FDR and Holm corrections for ties show the MAHRL and MADDPG methods significantly outperform PPO ($p < 0.01$). However, MAHRL is not significantly different from MADDPG ($p = 0.45$), but the distribution is skewed lower than MADDPG with more zero collision samples.

## 6.3 QUALITATIVE RESULTS

To evaluate the NC policies qualitatively, we show that agents learn to navigate arbitrary scenarios while avoiding collisions with both obstacles and other agents. First, we show a rasterized version of an example episode from the *Pursuit* environment in Figure 7 where agents learn to corner and tackle. Then, we show that agents can successfully and continuously navigate complicated environments of forced interactions, as seen in Figure 9a (Kapadia et al., 2011). Finally, robust clogging and trampling are shown in both low and high-density bottleneck egress scenarios, respectively Figure 9b.

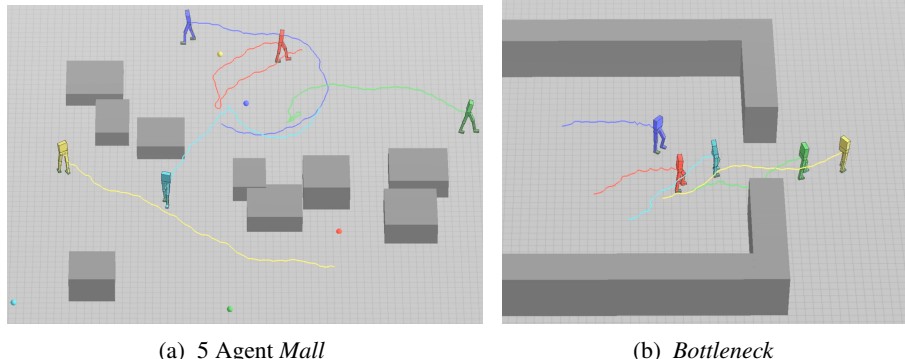

(a) 5 Agent *Mall*                    (b) *Bottleneck*

Figure 9: (a) Agents reaching series of targets in arbitrary environments (images in raster order). (b) Egress scenarios with a group of (left) 5 and (right) 21 agents. The density of the second group results in a physics-based bumping, pushing, and trampling.

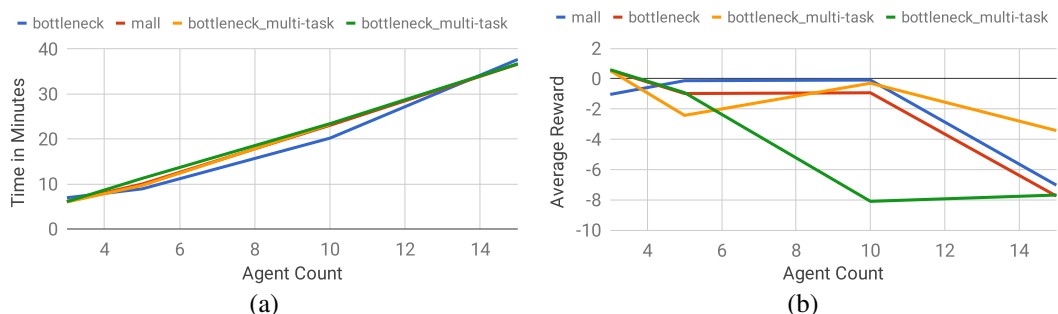

Figure 10: The performance of the method from two quantitative perspectives, (a) the computational performance with respect to agent count and (b) the generalization performance with respect to average reward value.

## 6.4 COMPUTATION AND GENERALIZATION

The deep integration of physical character control and distributed multi-agent navigation comes with a cost directly dependent on the number of active agents. In this section, we show two results in the same experiment, the computational costs of increasing the number of agents and the model generalization to more difficult scenarios. For two scenarios, mall and bottleneck, the number of agents is increased, and we record the average reward and computation time from the simulation. The agent-computation curve in Figure 10(a) indicates a linear trend in computational cost. While at agents counts in the 20s the simulation is not real-time, the most computationally expensive part is not the learning system but the physics simulation. Computationally efficient articulated body simulation is an active area of research (Erez et al., 2015). For accurate and stable simulation, we use a physics time-step of $0.0005$ s.

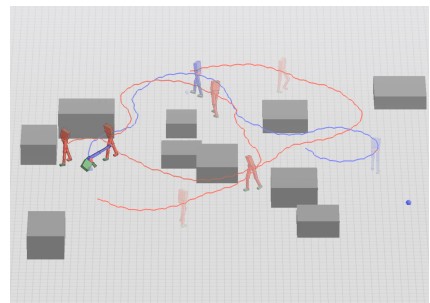

Figure 7: Rasterized overlays from the pursuit environment, where the pursuer agents (red) learn to work together to corner and tackle the navigating agent (blue). A video for this example can be found here.

To evaluate the learned policy's ability to generalize with group size, we vary the number of agents for select tasks after a homogeneous policy has been trained. The longer the policy can maintain a high average reward, the better the generalization. In addition, the average reward for two different types of policy training styles is compared. The first method trains on a single task at a time; the second method uses *multi-task* learning in hopes that a more generalizable task-independent structure is acquired. The *multi-task* method, often preferring to optimize easier tasks, does not appear to learn more robust

policies compared to the scenario-space based method (Kapadia et al., 2011). All generalization results can be seen in Figure 10(b). However, generalization remains a known and open issue of DRL methods (Zhang et al., 2018)

## 7 CONCLUSION

In this paper, we present a novel method where, for the first time, multi-agent navigation and physical character control are integrated. Multiple heterogeneous interacting agents can experience physical interactions, handle physical perturbations, and produce robust physical phenomena like falling and slipping. To achieve this, we developed an integrated model of MARL and HRL with *partial parameter sharing*. The evaluation of this approach shows how valuable it is for addressing the non-stationary learning problem of MARL in complex multi-agent scenarios. In particular, the heterogeneous *Pursuit* and *Soccer* scenarios learn complex routing and tackling behaviours akin to state-of-the-art competitive self-play approaches.

While our method produces promising results, the work is limited by the fixed LC partial parameter sharing. There is room for research in the area of training the LC and NC concurrently. For the NC, we introduced a set of reward functions to encourage human-like behaviour while navigating with other agents. The literature motivates these rewards, but balancing them is its own challenge. In the future, it may be beneficial to use additional data-driven imitation terms to encourage human-like paths. Finally, considerable effort was made to create combined locomotion, navigation, and behaviour controller that is robust to the number of agents in the simulation. However, robust generalization remains an open problem.

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

## 8 SUPPLEMENTARY MATERIAL

### 8.1 LOCOMOTION CONTROLLER

#### 8.1.1 REWARD

We found that a combination of imitation and end effector rewards leads to a robust and reactive motion. The goal is to reward joint angle targets which result in mimicking source motions. The

combination of these competing rewards leads to a complex balancing problem of many objectives–trading off imitation behaviour for footstep matching.

### 8.1.2 IMITATION

The imitation method is predicated on time indexed $0.5$ s long reference motions $q_i(t)$ created from motion capture data and augmented with footstep parameterized labels. The difference between the agent and the desired pose to imitate is computed as the weighted difference between the time indexed pose and the agent $r_{Lp} = \exp(-0.5 \sum_i ||\mathbf{s}^i_{char} - q^i(t)||^2)$. The motion database contains $7$ s of segmented motion that coarsely covers the space of possible footstep positions and the motions necessary to realize those footsteps. The accuracy and diversity of the Low-Level Controller (LLC)'s motion can be increased or made more robust with additional motion if desired, including stylized, disordered, running, and more.

However, if the agent matches these footstep goals too well, the smoothness of the motion will degrade. A balance is struck between learning robust footstep placement strategies and perfect mimicry–the reference motions and foot placements guide naturalness and should not be over-fitted.

### 8.1.3 END EFFECTOR

An additional reward is given to encourage the policy to match the end-effector positions in the imitation motion $r_{Le} = \exp(-0.15 \sum_e ||\mathbf{s}^e_{char} - q^e(t)||^2)$. Reward is also given for how well the agent matches the imitated motion's Centre of Mass (COM) $r_{Lcom} = \exp(-0.2 \sum ||\mathbf{s}^{com}_{char} - q^{com}(t)||^2)$

### 8.1.4 TORQUE PENALTY

To smooth the actions generated from the control policy, a reward for maximizing the above rewards with minimal torque cost is used $r_{L\tau} = \exp(-0.2 \sum_i \tau(\mathbf{s}^i_{char}))$. These torques are normalized by specified torque limits that keep the agent from displaying unrealistic strength.

### 8.2 TRAINING & SIMULATION ENVIRONMENT

The LLC state includes the proprioceptive-like joint information. In particular, the components of $\mathbf{s}_{char}$ consist of the COM relative distances and velocities of each links as well as the rotation and angular velocities of each link as shown in Figure 1(d). The trained policy actions are in the form of angle-axis targets for per-joint Proportional Derivative (PD) controllers. In this way, the action is used to set the desired position of each joint of the agent. This can be seen in Figure 1(e). The method is also able to learn policies for agents of different types. In Figure 8 we additionally show the method using a full humanoid LLC character.

The agent is simulated with the physics-based rigid body simulation environment Bullet Bullet (2015). The policy is trained in an on-policy fashion. At the beginning of each episode a goal $g_L$ is sampled from the simulation and updated every $0.5$ s. Episodes end when either a time limit $T$ is reached or the agent falls. Between training rounds $4096$ transition tuples are collected by simulating episodes in parallel. After enough tuples have been collected $16$ minibatch updates are performed with a batch size of $256$. The value function is updated $64$ with a minibatch size of $64$. For the policy and value function. The policy and value function are both modeled using a ANN with 2 hidden layers of size $512$ and $256$. The network uses ReLU activations through except for after the last layer where a *tanh* activation that outputs values between $[-1, 1]$ is used. The Adam stochastic gradient optimization method is used to train the ANN parameters Kingma & Ba (2014). The learning rates for the policy and value function are $0.0001$ and $0.001$ respectively. Details related to the learning algorithm can be found in the supplementary material.

Many previous methods have created robust controllers via imitation. These methods were not intended to be used in crowds or dynamic environments with potentially random disturbances from other interacting agents in the simulation. Given that we intend to use this agent in a physical multi-agent simulation where the agent may bump into other agents or obstacles when trained in a crowded setting, we apply additional methods to simulate pushes that may be encountered. That is, random pushes between $50 - 150$ N are applied every $3 - 5$ s for a duration of $0.1 - 0.3$ s to increase the robustness of the controller. Similarly, the motions in the motion database start from the same facing

direction. To make the agent more robust to distributed crowd scenarios, the agent's initial facing direction is randomized during training.

### 8.2.1 NC REWARD

Navigation combines the desire to move towards goals, while avoiding collisions with other objects, in an energy efficient way. The methods purpose is to implicitly learn low effort local turning behaviour while avoiding collisions without the need to explicitly describe its operation. To elicit this behaviour, a combination of reward signals are used. Primarily, a dense reward is used to encourage the agent to walk in the direction of its current goal.

$$r_{Hd} = \exp(-(\min(0, (u_{tar} * v_{com}) - v_{com}))^2) \qquad (4)$$

where $v_{com}$ is the agent's velocity and $u_{tar}^T$ is a normalized vector in the direction of the goal. For this work, a desired speed of $v_{com} = 1.0m/s$ is used.

A directional reward is not enough to encourage the agent to proactively reach its goal. To reinforce the importance of goal reaching behaviour, a large reward $r_{goal}$ for reaching the goal is added–being within $\sqrt{2}$ m. In this work, the value $r_{goal} = 20$ is used and comes from $horizon = 1/(1 - \gamma)$ and the maximum reward the agent can otherwise receive is $max(r_d) = 1$, a reward that will be more important than travelling to the goal should be at least $horizon * max(r_d)$. This reward greatly increases the goal reaching behaviour. However, it can have the unintended effect of making the agent seek its goal aggressively, by trampling other agents. To reduce this behaviour, a repulsive cost was added when the agent is within $\sqrt{3}$ m other agents

$$r_{Ha}(a) = \sum_{a' \in \{A-a\}} -(r_s + (l - dist(a, a'))) \qquad (5)$$

where $A$ is the set of agents in the simulation, and $dist(a, a')$ computes the Euclidean distance between the COM of agents $a$ and $a'$. We empirically found $r_s = 2.5$ and $l = 3.0$ work well for defining a distance dependent penalty in this case. This is intended to roughly approximate the power law of pedestrian interactions (Karamouzas et al., 2014). A similar repulsive cost is applied between the agent $a$ and obstacles $OB$ in the scene $r_{Hb}(a) = \sum_{ob \in OB} -2.5$ for each obstacle within 1m of $a$. The high reward for reaching the goal makes the agent very single-minded. The obstacle penalty is introduced to prevent the agents from using obstacles as affordances to regain balance.

Many RL simulation environments use a flag to indicate the episode or simulation end. This indicates that either the agent has reach its time limit or the agent has entered into an unrecoverable area of the state space, such as a fall. With multiple agents, the likelihood of a fall is high. It is not clear what to do when there are multiple agents being simulated. Terminating early, when one agent has fallen, is sub-optimal for other agents doing well, and waiting for every agent to fall wastes compute resources while most agents are collecting unhelpful data. We chose to reset the simulation when more than half the agents have fallen. However, fallen agents continue to act and need to be heavily penalized, so they receive a fixed reward of $-5$ which represents the largest penalty in our system.

The NC's final reward function is a combination of the task rewards and behavioural costs:

$$r_H(a) = \begin{cases} -5 & \text{if } fallen \\ r_{Hd} + r_{Ha}(a) + r_{Hb}(a) & \text{otherwise.} \end{cases} \qquad (6)$$

### 8.3 LEARNING DETAILS

The modification to this algorithm for the MARL method is to change line $7 - 8$ to execute an action for each agent in the simulation in parallel.

The Inputs are standardized wrt to all states seen so far. The rewards are divided by the variance. The variance is computed from a running computation during training that is updated after every data collection step. The advantage is batch normed.

**Hyper Parameter Exploration** Parameter exploration is a key process in acheiving the best results for DRL methods. In this work we explored different network architectures, policy variances, annealing the variance, learning rates and activation types. The results with the best performance are reported in the paper.

---

**Algorithm 1** Goal-Based Learning Algorithm

---

1: Randomly initialize model parameters $\theta_\pi$ and $\theta_v$
2: **while** not done **do**
3:     **for** $i \in \{0, \ldots N\}$ **do**
4:         $\tau_i \leftarrow \{\}$
5:         **for** $t \in \{0, \ldots, T\}$ **do**
6:             $a_t \leftarrow \pi(\cdot|s_t, \theta_\pi)$
7:             $s_{t+1}, r_t \leftarrow$ execute $a_t$ in environment
8:             $\tau_{i,t} \leftarrow < s_t, a_t, r_t >$
9:             $s_t \leftarrow s_{t+1}$
10:         **end for**
11:     **end for**
12:     Update value function $V_\pi(\cdot)$ parameters $\theta_v$ using $\{\tau_0, \ldots, \tau_N\}$
13:     Update policy parameters $\theta_\pi$ using $\{\tau_0, \ldots, \tau_N\}$
14: **end while**

---

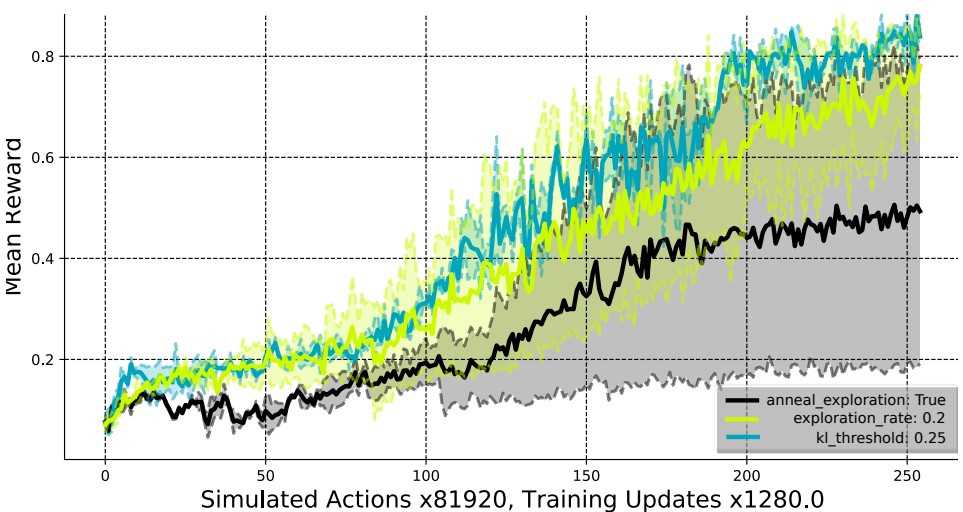

Figure 11: Hyperparameter exploration learning curves for the LC training.

## 8.4 RESULTS DETAILS

### 8.4.1 LC HYPERPARAMETER EXPLORATION

The LC hyperparameter exploration learning curves in Figure 11 illustrate the process of finding optimal training parameters. It was found that annealing the policy variance over time did not increase the learning efficiency. As well, the best settings for the policy variance and *kl_threshold* are 0.2 and 0.25 respectively.

### 8.4.2 MULTI-AGENT NAVIGATION UNIT TASKS

In this section, we show several multi-agent navigation tasks and their qualitative performance using our method. These are rendered as birds-eye views of scenarios with each agents trajectory over time rendered. Each agents goal is rendered as a red point within a red circle and the shortest linear path to that from the agent's current position is rendered as a red line. We show that agent's learn to navigate towards arbitrary goals repeatedly hitting its mark with each new goal in Figure 12. Predictive reciprocal collision avoidance is important in multi-agent navigation and lends naturalness to qualitative results. We show our method learns high value reipcrocal collision avoidane strategies, and we overlay the sample points for the velocity state space to illustrate how this is acheived in Figure 13. We show that this generalizes to the introduction of obstacles in both Figure 14 & 15.

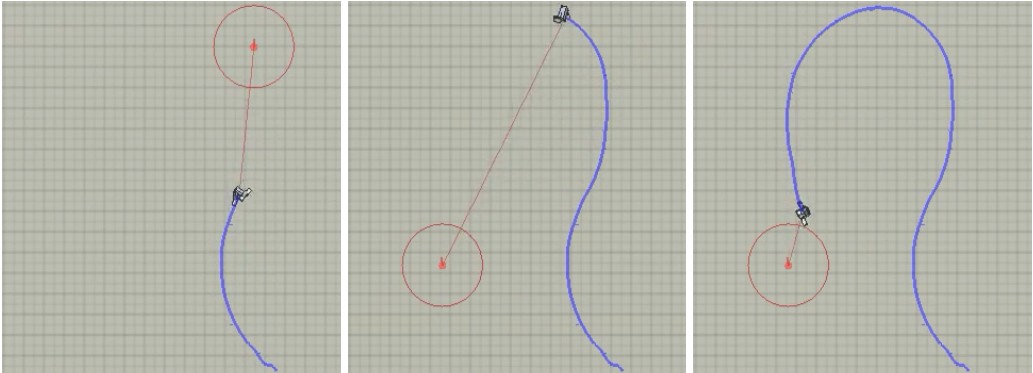

Figure 12: The agent moves towards and reaches goals repeatedly (top-left to bottom-right). The learned model produces smooth goal seeking trajectories show in in blue.

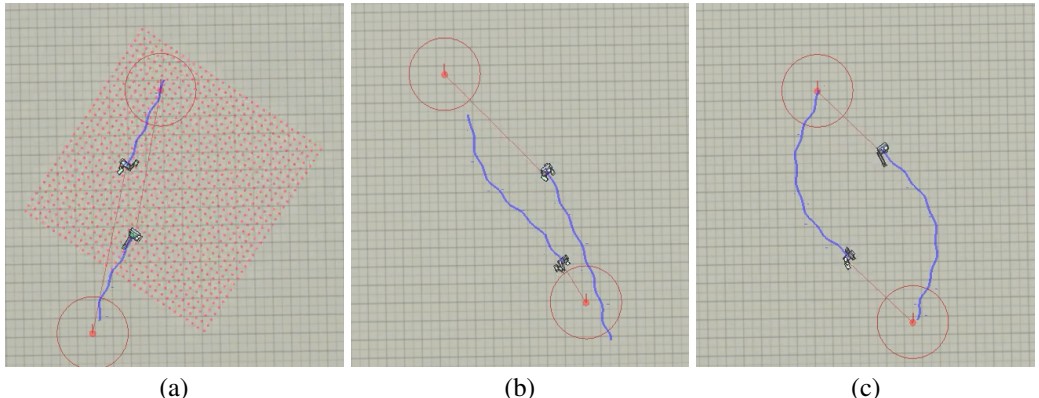

|  (a)  |  (b)  |  (c)  |

Figure 13: Three examples demonstrating reciprocal collision avoidance. The size and position of the egocentric state sampling field relative of the agent is shown in (a).

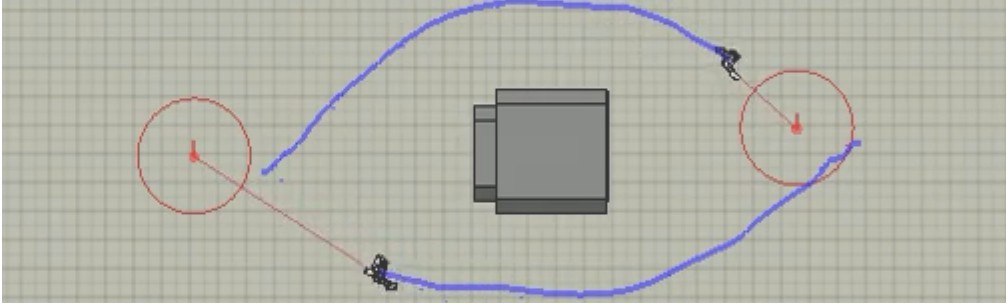

Figure 14: Reciprocal collision avoidance with obstacles. Each agent's initial position is the target location of the other agent.

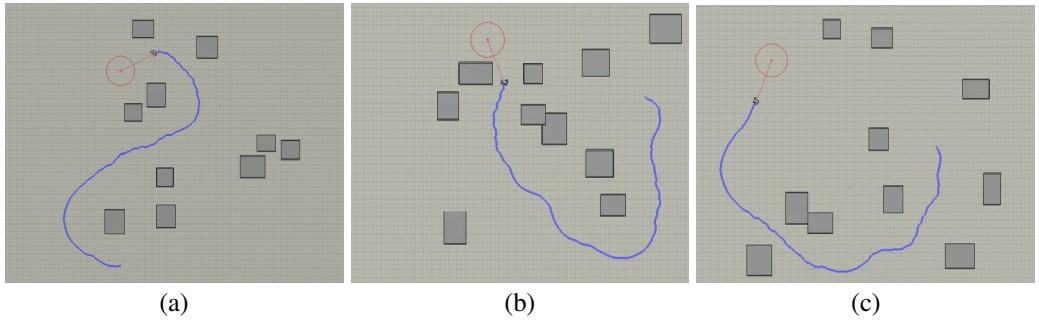

(a)                              (b)                              (c)

Figure 15: An agent reaching goals in arbitrary complex environments. The red circle indicates the final goal.

## 8.5 ADDITIONAL DISCUSSION

### 8.5.1 LC AGILITY

While often in the crowd simulation literature it is common to have agents that can turn on point or produce holonomic motions, this is unrealistic for articulated agents. Interestingly, the agent learned stopping and in-place turning behaviour which was not contained in the imitation data the LC was trained on, indicating the system can generate behaviours beyond its design. However, the agents in this work do not make unrealistic sharp turns. This is in part related to the NC being able to avoid falls, however there are many other factors: The LC motion capture data, $g_L$ selection during LC training and that the LC was not trained with other agents. Progress in any of these areas can improve the responsiveness of the LC and is left for future work.

While the LC is goal driven, this goal is only based on foot placement, a fully interactive agent can have many other types of short term goals, including where to put one's hands to manipulate items in the environment, like doors.

