# OpenReview forum: "Multi-Agent Hierarchical Reinforcement Learning for Humanoid Navigation"
_ICLR.cc/2020/Conference — Reject_

### Official Review · AnonReviewer2 · 2019-10-15
**Official Blind Review #2**

**Rating:** 3

**Review:**

Summary:
This paper looks at the MARL problem in high-dimensional continuous control settings. To improve learning in this multi-agent setting, they propose to pre-train a lower-level policy that takes as input foot-step goals and is executed for a fixed number of timestep, thereby simplifying both the learning and exploration.

I'm a bit unsure of how to evaluate this paper. On the one hand, I believe it has several contributions:
- Proposing a new MARL - continuous control environment
- Proposing a new lower-level policy for high-demensional continuous control environments, including how to learn it
- Using it to perform MARL in this environment

On the other hand, it is hard to say what the _main_ contribution is, which in turn makes it difficult to evaluate whether the experimental evaluation is sufficient:
Clearly, a main part of the paper is the work done to construct the hierarchical setup, including goal space, observation space and reward functions. However, this work, as far as I can tell, is separate from the MARL problem. Furthermore, there are several similar ideas already published, so comparison against those (for example by J. Peng, N. Heess or J. Mere) either as argument or even better as experiment, would be helpful to evaluate the quality of the proposed hierarchy.

On the other hand, there is the application of the hierarchical setup to the MARL problem. However, as far as I can tell, there is no difference between applying such a hierarchy to the MARL case and to the single agent problem. Especially if the lower-level component of the hierarchy is pre-trained in a non-MARL setup, it can just be seen as part of the environment from the point of view of the MARL training, offerring limited new insight into MARL.
I believe in the second paragraph of 4.1 the authors provide some insight into this matter, however, I have to admit I do not understand this paragraph:
- Why does temporal correlation reduce the non-stationarity of the MARL problem?
- Why does structured exploration reduce the number of network parameters that need to be learned?
- Why does partial parameter sharing make it easier for each agent to estimate other agents potential changes in behavior?


In summary, I think this is interesting work, but a clearer explanation of the relationship between HRL and MARL, as well as a clearer main argument, supported by experimental evidence, would greatly improve this paper.

Edit:
Thank you for your response.

Unfortunately, I don't feel like it sufficiently addresses my questions and concerns.
I do apologize if my original comment wasn't clear regarding the contribution part of the paper. What I was trying to say is not that I didn't see the individual contributions of the paper, but instead that the paper does multiple things simultaneously, without comparing against the relevant baselines for any of the individual contributions.

Regarding my questions: I understand where the temporal correlation is coming from in an HRL setting. However, what was not clear to me is how this reduces the non-stationarity of MARL.
I also understand that HRL can reduce the number of parameters, but I don't see how structured exploration reduces the number of parameters.
And lastly, I also can see how parameter sharing can simplify the learning, but I still don't see how it would allow agents to estimate the behaviour change of other agents easier.
I feel like in the paragraph in questions, a lot of causes and effects are mixed up and more careful descriptions of the benefits of the algorithm would help.

I want to re-iterate that I think that the submitted work by the authors is impressive and can provide valuable insights, but I believe it requires more work and more relevant baselines.

**Experience Assessment:**

I have read many papers in this area.

**Review Assessment: Checking Correctness Of Derivations And Theory:**

N/A

**Review Assessment: Checking Correctness Of Experiments:**

I assessed the sensibility of the experiments.

**Review Assessment: Thoroughness In Paper Reading:**

I read the paper at least twice and used my best judgement in assessing the paper.

---

> ### Author Response · Authors · 2019-11-14
> **Discussion and clerification on method**
>
> We are grateful for your time and comment on the work. We start by further explaining the contributions in the paper. Our main contribution is the combination of MARL with HRL to enable the decentralized learning of controllers that can navigate and seek goals in a robotic humanoid simulation. The unique combination of methods allows us to learn these sophisticated controllers with far less data than methods without hierarchy (cite openAI Emergent Tool Use from Multi-Agent Interaction). Second, we also consider the environment in the paper another contribution. Few multi-agent environments simulate dynamics, and none have articulated humanoid robots that observer their world using egocentric vision. We plan to release this environment with the work to allow other researchers to pursue and make progress on important complex tasks.
>
> Many multi-agent problems have been studies using simple robot models (point-mass, etc), where more complex and realistic models have used the problem because significantly more challenging. However, often, an assumption can be made that the robots in the environment share similar morphology. We propose a method that uses a form of goal-conditioned RL to learn task agnostic low-level policies that can simplify the share control structure across robots. In most HRL methods, the lower level can be viewed as part of the environment, yet this restructuring of the environment enables faster and more capable learning.
>
>
> Here we clarify some of the proposed advantages of the method. First, the use of HRL enables temporal correlation in action exploration that helps reduce the non-stationarity challenge. The advantage of this temporal correlation is shown empirically in Figures 2 and 3 where the PPO policies do not improve on learning the tasks. This property can be understood to reduce the variance in the policy gradient. Instead of having the policy sample an action every step instead, the low-level policy is triggered for $k$ timesteps with a goal proposal. For these $k$ timesteps, no noise is added to the low-level policy outputs. Similarly, this $k$-step structured exploration enables learning. If we think of the policy as a type of VAE that is learning an encoder (high-level) that is trying to learn a good latent goal (z) that will result in the low-level performing the desired sequence of actions. The HRL structure is reducing the dimensionality of the control problem given a low-level designed to perform diverse behaviour wrt to the goal (cite Heess and DIAYN). Last, the partial parameter sharing appears to make the learning problem easier. We know it is challenging to learn Q functions, which implies that the centralized methods that use Q functions will not scale well. We compare our method to MADDPG, a popular centralized method that works by treating the problem as a single agent problem with complete information. In our case, this method results in a significant increase in network parameters for the Q function, which leads to poor learning performance, as can be seen in Figures 2 and 3. Our particular configuration allows our method to be decentralized, making the individual network for each agent more straightforward. We are also interested in generalization to different numbers of agents after training, which is also problematic for centralized methods. In short, decentralized learning will allow for more general methods, and HRL enables the learning of sophisticated controllers.

---

### Official Review · AnonReviewer1 · 2019-10-22
**Official Blind Review #1**

**Rating:** 3

**Review:**

The submission proposes a method for hierarchical RL in multiagent settings. In particular it proposes to explicitly decouple training of a high-level and low-level controller with grounded the controller interface as goals in the environment to reach for the low-level controller. The model is trained via PPO with GAE and evaluated on a small set of multi agent locomotion tasks.

The paper is overall well written and intuitive but limited in evaluation and novelty (see e.g. [1,2] ) with only limited modifications (sharing low-level controller) for the multi agent case. Furthermore, the experimental section does not compare to other forms of hierarchical approaches for MARL, and generally only provides a single comparison to PPO & MADDPG. To evaluate the impact of the proposed changes in this paper, one would have to perform extended evaluations and ablations for the submission.

A large part of making the MA system work well is based on reward shaping which nearly fills all of page 5. This is clearly interested in as far as solving this particular task but does not provide any general insights for the design of (MA)RL algorithms.

The experimental section includes various mistakes (see under minor) and misses to describe figures, leading to the assumption that additional time is required for a more detailed evaluation of the algorithm (including more domains and in particular baselines).
Regarding the challenges (and focus on learning simple tasks), reference [3] might be of interest to the authors.

Minor
- Direct duplication of text between parts of section 5.3 and 8.3 leading to the duplication of the error of describing the value function learning rate as 0.000.
- Self-referential sentences in the supplementary materials (i.e. referral to itself)
- Missing references on page 3
- The egocentric velocity field is not described (section 5)
- Section 3.1: maximize
- The wording new paradigm in MARL might be unsuited given existing work on complex domains.
‘Our proposed approach represents the first physics-based simulation of its kind that supports MARL.’ This sentence remains unclear as the authors do not propose a simulation engine.
- Text on experiment figures is much too small.

[1] Andrew Levy, Robert Platt, and Kate Saenko. Learning Multi-Level Hierarchies with Hindsight. In International Conference on Learning Representations, 2019.

[2] Ofir Nachum, Shixiang Shane Gu, Honglak Lee, and Sergey Levine. Data-efficient Hierarchical Reinforcement Learning. In Advances in Neural Information Processing Systems, pp. 3303–3313, 2018.

[3] Ray Interference: a Source of Plateaus in Deep Reinforcement Learning Tom Schaul, Diana Borsa, Joseph Modayil and Razvan Pascanu


**Experience Assessment:**

I have published one or two papers in this area.

**Review Assessment: Checking Correctness Of Derivations And Theory:**

I assessed the sensibility of the derivations and theory.

**Review Assessment: Checking Correctness Of Experiments:**

I carefully checked the experiments.

**Review Assessment: Thoroughness In Paper Reading:**

I read the paper at least twice and used my best judgement in assessing the paper.

---

> ### Author Response · Authors · 2019-11-14
> **Extended analsysis and baseline comparison**
>
> We thank the reviewer for their time and comments on the work. Concerning including more comparison and ablations in the paper, we have performed an extended analysis of our method to the baselines across many environments. See Figures 2,3,5 for more learning curve results and baseline comparisons and Figure 6 for qualitative metric analysis. We show that our method outperforms the baselines across multiple environments. In the paper, we include many details on the environment rewards and design as we consider these simulation tasks part of the contribution of the work. The simulation tasks contain robotic humanoid characters that need to learn how to navigate given egocentric vision. No other simulation is available that combines these challenges. The simulation will be released with the work for others to use and build on multi-agent learning methods.
>
> We have reviewed the provided references and have included them in the paper.

---

### Official Review · AnonReviewer3 · 2019-10-23
**Official Blind Review #3**

**Rating:** 3

**Review:**

This paper proposes a multi-agent hierarchical reinforcement learning algorithm so that multiple humanoid robots can navigate in multi-agent settings (e.g. avoid collisions, collaboration, chase and escape) in a physically simulated environment. The key difference of this paper with the prior work on MARL is that it used an accurate physics simulation of humanoid robots. This is the main reason of using the hierarchical RL.

In general, I like this paper. It is an important step towards multi-agent learning in complex physical environments. The results look appealing, too. However, I voted for "Weak Reject" for two reasons. First, the technical contribution is lean. Neither the multi-agent learning or the hierarchical learning of the algorithm is novel. The combination of these two methods seems straightforward. Once a low-level walking controller is trained, the high-level multi-agent navigation control is not much different from simple environments, e.g. point mass control, used in the previous works. I do not understand the "deep integration of MARL and HRL" that is claimed in the Introduction. I also do not agree with another claim that "We consider the simulation and training environment to be another novel contribution... few simulator support more than one agent, at most 2". In most of the simulators that I am familiar with, such as Mujoco, Bullet, DART, it is straightforward to add multiple simulated robots.

Second, the writing can be greatly improved. Almost half of the technical details are buried in "8. Supplementary material". Since it is not fair to use "Supplementary material" as a way to extend the page limit, I will make my judgement of the paper solely based on the contents up to Section 7. In the main text (up to Section 7), there is no mentioning of how the low-level controllers are learned, and how to combine PPO in a MARL partial parameter sharing setting. I think that these are important details and may also be the contributions of this paper. Most of these should be moved to the main text.

Here are some more suggestions on writing:
1) Certain paragraphs in the main text can be significantly shortened, such as the reward shaping in Section 5.2.
2) It would be great if the paper can clearly define the experiments: "waypoint", "oncoming", "mall", and "bottleneck".
3) The paper needs a thorough proof-reading. There are many grammar mistakes, typos, missing citations. For example,
promiss->promise
week signal->weak signal
missing citation [?] in page 3
reuse the same symbol v_{com} for agent's velocity and desired speed in eq(3)


**Experience Assessment:**

I have read many papers in this area.

**Review Assessment: Checking Correctness Of Derivations And Theory:**

I carefully checked the derivations and theory.

**Review Assessment: Checking Correctness Of Experiments:**

I carefully checked the experiments.

**Review Assessment: Thoroughness In Paper Reading:**

I read the paper at least twice and used my best judgement in assessing the paper.

---

> ### Author Response · Authors · 2019-11-14
> **Clerify contribution and add soccer task**
>
> We appreciate your time and comments on the work. While the method is the first to be applied to a multi-agent simulation with articulated humanoid character, our main contribution is a method to allow sophisticated controllers to be learned in this case. Our unique combination of structured learning enables the learning of strong polices without incredible amounts of computing time (cite openAI Emergent Tool Use from Multi-Agent Interaction). We also argue that the learning and control problem for the high-level policies is more complicated than a “point mass” environment. The high level needs to learn strategies to cope with a dynamic simulation that includes, pushes, slips, balancing, etc, all through the capabilities of a low-level policy while optimizing a goal-seeking objective. This simulation environment it also novel in that no other simulation has put multiple dynamic humanoid agents in a simulation that observe each other using egocentric vision. In other simulation libraries, it is possible to add more agents, but no tasks have been constructed or learned that match the complexity in this work.
>
> We agree that the paper writing can be improved. Significant edits to the paper have been made to make the method and its contribution more clear. These edits include moving the details for training the goal-conditioned low-level controller to the main paper, including adding a new task for 2-on-2 soccer for which MAHRL has shown significant progress on learning.

---

### Decision · Program_Chairs · 2019-12-19

**Decision:**

Reject

**Comment:**

This paper presents an approach combining multi-agent with hierarchical RL in a custom-made simulated humanoid robotics setting.

Although it is an interesting premise and has a compelling motivation (multi-agent, real-world interaction, humanoid robotics), the reviewers had some trouble pinpointing what the significant contributions are. Partly this is due to lack of clarity in the presentation, such as with overlong sections (eg 5.2), unclear descriptions, mistakes in the text, etc. Reviewers also remarked that this paper might be trying to do too much, without performing the necessary experiments/comparisons and analyses needed to interpret the contributions of each component.

This work is definitely promising and has the potential to make a nice contribution, given some additional care (experiments, analyses) and rewriting/polishing. As it is, it’s probably a bit premature for publication at ICLR.